# Comparison of Fissure Sealant Chair Time and Patients’ Preference Using Three Different Isolation Techniques

**DOI:** 10.3390/children8060444

**Published:** 2021-05-25

**Authors:** Rahif E. Mattar, Ayman M. Sulimany, Saad S. Binsaleh, Ibrahim M. Al-Majed

**Affiliations:** Department of Pediatric Dentistry and Orthodontics, College of Dentistry, King Saud University, Riyadh 11545, Saudi Arabia; asulimany@ksu.edu.sa (A.M.S.); ssbinsaleh@ksu.edu.sa (S.S.B.); ialmajed@ksu.edu.sa (I.M.A.-M.)

**Keywords:** children, fissure sealants, Isolite, isolation, patient preference, time, permanent molars, randomized clinical trial

## Abstract

This randomized clinical trial aimed to evaluate the patient’s preference and chair time needed during pit and fissure sealant placement under three isolation techniques (Isolite system, rubber dam isolation, and cotton roll isolation). Participants, aged 6–15 years and requiring four sealants on the first or second permanent molars, attending the pediatric dental clinics at King Saud University in Saudi Arabia were enrolled according to the inclusion criteria. Each participant received sealants on three random first or second permanent molars using three isolation techniques. The time required for sealant placement was recorded for each technique. Following sealant placement, an interview-based questionnaire was administered to the participants to evaluate their preference regarding the isolation techniques. Forty-eight children (23 male and 25 female) with a mean age of 8.58 ± 1.93 years participated in this study. The mean chair times were 248.14, 255.89, and 243.29 s for the Isolite system, rubber dam isolation, and cotton roll isolation, respectively. Approximately 79% of participants considered cotton roll isolation to be the most comfortable, whereas approximately 71% were significantly less likely to use rubber dam isolation again. In conclusion, there were no significant differences in sealant placement time among the three isolation techniques. However, cotton roll isolation was the technique that was most preferred by the participants.

## 1. Introduction

Dental caries represent one of the most common chronic and multifactorial diseases affecting the human population [1]. Eighty to ninety percent of dental caries are present in the pits and fissures of the permanent teeth in pediatric patients [2]. Several epidemiological studies have indicated that caries in children are considered particularly critical and that the incidence of dental disease among children in Saudi Arabia is increasing [3]. The prevalence of dental caries is considered high, affecting 69%, 73%, 62%, and 85% in Riyadh city, Dammam city, Makkah city, and Abha city, respectively [4,5,6,7]. Thus, there is a need to limit pathogen growth and caries activity by prevention rather than cure [3]. Sealants have been widely discussed since their development due to their ability to inhibit the growth of bacteria by eliminating food particle accumulation in these fissures [8,9]. The success of pit and fissure sealants (PFS) depends on placing the sealant in an ideal situation [10]. Effective isolation from moisture is a key factor in the success of sealants [11], and several methods have been invented and studied to assess the long-term retention rates of sealants, such as rubber dam isolation (RDI) [12] and cotton roll isolation (CRI) [13]. According to the American Dental Association, RDI is the recommended technique for placing fissure sealants [12], while CRI is the most common technique used by pediatric dentists [13]. A common difficulty found with RDI is children’s negative reaction to this method [14,15]. CRI has been reported to cause gagging due to the placement of cotton rolls on the lateral portion of the tongue or as a result of insufficient isolation, causing the child to experience the taste of the materials and a constant need to replace the moistened cotton rolls [15]. Recently, a new isolation system called the Isolite system (IS) was introduced. The Isolite system is an illuminated dental isolation system that is intended to isolate two quadrants simultaneously. The isolation system is connected with a silicone latex-free mouthpiece, which comes in various sizes. The mouthpieces have four main characteristics: continuous suction, retraction of soft tissue, acting as a bite block, and intraoral LED illumination [16]. Limited studies have assessed the patient’s acceptance and PFS chair time of IS in comparison to CRI [17] and RDI [18]. Collette et al. [17] reported that IS significantly reduced sealant chair time when compared to CRI; however, there was an insignificant difference in patient preference. Alhareky et al. [18] showed a significant difference in both chair time and patient preference for IS in comparison to RDI. To the extent of our knowledge, there are no clinical studies comparing patients’ preference, time efficiency, and pit and fissure sealant retention using IS compared to RDI and CRI in a single randomized clinical trial utilizing the four-handed delivery system. This study was intended to determine the difference, if any, in chair time when placing PFS using IS compared to RDI and CRI and to assess patient preferences of the isolation techniques. Pit and fissure sealants placed in this study will be clinically followed up in the future to assess the retention rates of the three isolation techniques.

## 2. Materials and Methods

This split-mouth randomized clinical trial was conducted between November 2018 and January 2020 at the College of Dentistry at King Saud University, Riyadh, Saudi Arabia. The study followed Consolidated Standards of Reporting Trials (CONSORT) recommendations (Figure 1). This study was registered at the ISRCTN with study ID ISRCTN10452417 and was approved by The Institutional Review Board (E-18-3376) and by the College of Dentistry Research Center (PR 0103) at King Saud University Riyadh, Saudi Arabia.

### 2.1. Sample Size

Power analysis for a repeated-measures ANOVA to compare the statistical significance of mean chair time using the three different techniques (Isolite system, rubber dam isolation, and cotton roll isolation) to provide 3 fissure sealants was conducted in G*Power software. To determine the required sample size, an alpha of 0.05, a power of 0.80 (80%), and a medium effect size (f = 0.30) [19] were used. Based on these assumptions, the desired sample size was 32. Additionally, to compare the statistical significance of the proportion of patient preference among the three techniques, assuming a higher preference of the Isolite system and with an effect size (w = 0.50) with 2 df, an alpha of 0.05, and a power of 0.80 (80%), the required sample size was 40. For an anticipated attrition rate of 20%, *n* = 8 was added, and the required sample size was 48 participants.

### 2.2. Inclusion and Exclusion Criteria

Forty-eight children aged 6–15 years attending the pediatric dental clinics were enrolled in the study according to the inclusion criteria: (1) normal healthy patient with ASA 1 according to the American Society of Anesthesiologists Classification; (2) children willing to participate in the study; (3) legal guardian consents to the participation of the child in the study; (4) participant must require 4 pits and fissure sealants on all first or second permanent molars with an International Caries Detection and Assessment System (ICDAS) score of 0–2 [20]; (5) children speak Arabic or English. The exclusion criteria were (1) special needs children; (2) conditions requiring emergency dental treatment (abscess, draining sinus, cellulitis); (3) partially erupted permanent first or second molars; (4) enamel/dentin anomalies; (5) uncooperative children, with Frankl Behavior Rating Scale of 1 or 2; (6) children with a severe gagging reflex; (7) children allergic to latex; (8) children with recent dental radiographs taken but not available on the computer system.

### 2.3. Screening Participants

Each participant was screened by a single operator (R.E.M.) to determine their eligibility for the study. Medical history and demographic information were collected, and a thorough clinical examination of the first and/or second permanent molars was achieved using a mouth mirror and a blunt instrument to evaluate the patients against the inclusion/exclusion criteria. Right and left bitewing radiographs were taken for each participant to confirm the complete absence of dental caries or dental anomalies in the first or second permanent molars. The purpose of the study was explained to the child and their guardian/parent, and informed consent and assent were obtained.

### 2.4. Randomization

Random numbers were generated using a computer program (MedCalc), and all eligible study participants were randomized by using simple block random allocation to ensure a balanced randomization for each isolation system. Numbers were printed on separate papers, folded, and placed in opaque sealed envelopes. Each participant chose an envelope and was assigned to the printed participant number. Each number identified the teeth included in the study, the isolation technique specified for each tooth, and the sequence of isolation application to follow.

### 2.5. Intervention

Each participant received three PFS on their permanent first or second molars using three different isolation techniques, according to randomization by a single operator (R.E.M.). The fourth molar that was not included in the study received PFS after completing the three molars enrolled in the study and after interviewing the participants. The isolation system used for placement of the fourth PFS was chosen according to the participant’s preference.

#### 2.5.1. Prophylaxis

Prior to isolating the enrolled teeth and placing the sealants, all four molars were cleaned by the operator with pumice using a low-speed handpiece and a prophylaxis polishing brush (Dentamerica^®^, Taipei, Taiwan). The teeth were rinsed thoroughly while using a high-volume suction held by a dental assistant; later, the teeth were dried using a dental air spray. Finally, a blunt instrument was used to check for any remaining debris in cases where debris was present, and recleaning was performed if necessary.

#### 2.5.2. Rubber Dam Isolation Procedure

The soft tissue around the chosen tooth was dried using a dental air spray. A topical anesthetic agent of 20% benzocaine gel (Sky-Caine^®^ Gel, Skydent Inc., Manhattan, NY, USA) was applied using a disposable dental Q-tip for 10–20 s according to the manufacturer’s instructions. A medium gauge 6″ × 6″ latex powder-free rubber dam (Dental Dam, Henry Schein Inc., Melville, NY, USA) was punched according to the specific tooth number and quadrant. A 14- or 14A-sized clamp was flossed, held in the clamp holder, and placed on the specific tooth to ensure that it was secure, without any rocking by pushing the bow of the clamp with an index finger. In the case of rocking, the clamp was readjusted. The rubber dam sheet was then held in both hands of the operator, stretching out the punched hole and sliding it over the bow of the clamp and adapting it on the tooth. The frame was put in place, and sealing was verified. A high-volume suction was placed by a dental assistant.

#### 2.5.3. Cotton Roll Isolation Procedure

Medium size #2 cotton roll (Cotton Roll, Jaan International Training Co., Ltd., Guangzhou, China) was applied and secured by the operator. For maxillary isolation, cotton rolls were placed on the cheek side of the teeth in the mucobuccal fold. For the mandible, cotton rolls were placed in both the mucobuccal fold and the lingual side of the arch. Cotton rolls were replaced in cases of extreme moisture and after enamel etching. A high-volume suction was placed by a dental assistant.

#### 2.5.4. Isolite System Procedure

An appropriate disposable Isolite mouthpiece size (Isolite Systems^®^, Zyris, Inc., Santa Barbara, CA, USA) was chosen by measuring the participants’ interincisal opening by the finger method [16]. The participants lips were lubricated with petroleum jelly. The isthmus, which is considered the narrowest area in the middle of the Isolite mouthpiece, was placed at the corner of the mouth, and the participant was instructed to open his/her mouth wide. The mouthpiece was inserted into the participant’s mouth while folding the cheek shield component forward toward the tongue retractor. The Isolite mouthpiece included a bite block component that was placed on the occlusal surface, and the participant was instructed to bite. Finally, the cheek shield was adjusted into the buccal vestibule, and the tongue retractor was adjusted into the tongue vestibule (Figure 2). This was all in accordance with the manufacturer’s instructions.

#### 2.5.5. Pit and Fissure Sealants Procedure

After applying each isolation technique, the participant was given a mirror to see each isolation technique in his/her oral cavity for recognition during the interview-based questionnaire. The mirror was handed to the participants by an assistant immediately after the etchant was applied, and it was removed prior to light curing. Each tooth was etched with 38% phosphoric acid dental etching gel (Etch-Rite™, PulpDent, Watertown, MA, USA) for 20 s [21,22]. The etched tooth was rinsed thoroughly to remove all organic particles from the micropores and dried for 15 s each. After drying, examination of the tooth was performed to ensure a properly etched enamel with a frosty white appearance [23]. Pit and fissure sealant (Clinpro™ Sealant, 3M ESPE, St. Paul, MN, USA) was placed on the tooth surfaces. For the maxillary molars, the occlusal and lingual surfaces were sealed. For mandibular molars, the occlusal and buccal surfaces were sealed. If any air bubbles were present, they were removed by a sharp explorer or a micro brush. The sealant was cured for 20 s according to the sealant manufacturer. Finally, the occlusion was checked, and adjustments were carried out using finishing burs if needed. A single operator performed all PFS placements with the aid of a dental assistant (four-handed delivery system).

#### 2.5.6. Time Measurement

The time was measured in seconds by an assistant using a stopwatch for all three isolation techniques. In cases of RDI, the beginning of topical anesthesia placement was the beginning of the time measurement. The placement of the first cotton roll was used for CRI, and the interincisal measurement was considered the beginning of the time measurement for IS. For each system, the end of the measurement was when the isolation system was completely removed from the participant’s mouth.

#### 2.5.7. Interview-Based Questionnaire

After completing all three PFSs, a 10-item interview-based questionnaire was administered to measure the participant’s acceptance of the different isolation techniques. This was carried out by interviewing and showing the participant three printed pictures of the three isolation techniques in the oral cavity in order to facilitate choosing the appropriate answer. The English questionnaire was designed and then validated by pediatric dentist experts. Following IRB approval, a pilot study was conducted for questionnaire validation. The questionnaire was forward and backward translated into Arabic for Arabic speakers.

### 2.6. Statistical Analysis

Repeated measures ANOVA was used to compare the time taken to place sealants among the groups, and the potential association between the age and time taken was tested using Pearson’s correlation coefficient. The time needed for sealant placement for each technique was further analyzed by arch, sex, and tooth type using independent *t*-tests. The differences in the preference of material for each of the patient preference questions were tested using the one-sample chi-square test. The level of significance for all tests was set at *p* < 0.05. All data analysis was performed using SPSS ver. 25 data processing software (IBM-SPSS, Armonk, NY, USA).

## 3. Results

### 3.1. Demographic Data

The sample comprised 48 (23 male, 25 female) children aged between 6 years and 15 years (mean age = 8.58 years, SD +/− 1.93 years). Although the boys were slightly older (mean age = 8.69 years, SD +/− 2.16 years) than the girls (mean age = 8.48 years, SD +/− 1.73 years), the difference was not statistically significant (*p* = 0.704). Sealants were placed on 144 teeth, of which the majority were maxillary first molars, and no statistically significant differences in tooth type among the groups was found.

### 3.2. Overall Time Analysis

The overall time taken for each type of isolation technique plus sealant placement was compared. It was observed that cotton rolls took the least amount of time, while rubber dams took the longest amount of time (Table 1). However, among the groups, it was observed that there was no significant difference in the time of sealant placement between methods of isolation (F = 0.151, *p* = 0.700).

### 3.3. Time Compared between Arches

When the time for isolation was compared between the arches, it was observed that the placement of sealants in teeth isolated with either Isolite or rubber dams took longer in the maxilla than in the mandible. On the other hand, cotton rolls took less time in the maxilla than in the mandible (Table 2). However, none of these findings reached the level of significance at *p* < 0.05.

### 3.4. Time Correlated with Age

When the overall time taken for each type of sealant placement was correlated with age, a negative association was seen for all three types of isolation. This suggests that the placement time decreases as children grow older. This association was the strongest in the Isolite group and the weakest in the cotton roll group. However, the associations were not statistically significant (Table 3).

### 3.5. Patient Preference

Participants were asked about isolation noise, least preferred, and future preference. Isolite was regarded by the participants as the noisiest of the isolation techniques (*n* = 32). The RDI (*n* = 34) was significantly less likely to be used again than the IS (*n* = 13) and CRI (*n* = 1). Finally, when patient preference for isolation was recorded, it was noticed that the patients preferred the CRI to both RDI and IS. These differences were statistically significant in all of the categories explored (Figure 3).

## 4. Discussion

This randomized clinical study was intended to detect the time difference of PFS application with three isolation techniques. Additionally, the study investigated the participants’ preferences regarding noise, comfort, and future choice of isolation. The results indicated that although more time was required to apply PFS using RDI and less time was required with CRI, the differences were insignificant. These findings differ from those of Collette et al. [17] and Alhareky et al. [18], where CRI and RDI were found to have a significantly increased time of sealant placement in comparison to IS. This difference may be due to excluding the process of molar prophylaxis from the recording time of this study, unlike previous studies. Prophylaxis was excluded from our recording because of the difficulty in standardizing the cleaning time for each tooth when the amount of plaque and food particles on each molar can vary significantly, hence requiring more or less cleaning time.

It is also possible that the insignificant difference was associated with the starting point of the recording, hence affecting the results. Collette’s starting time for recording for IS was at the beginning of IS insertion, and Alhareky’s was when adapting the IS mouthpiece on the system, including any adjustments needed, whereas in this study, the interincisal measurement was considered the beginning of the time recording. The authors believe that interincisal measurement is a vital step to be included, since choosing the correct mouthpiece size results in effective moisture control and allows for a successful PFS, which is similar in importance to placing the topical anesthesia prior to RDI to ease clamp discomfort; both are believed to be inevitable and essential part of the process. It was also found that operating with a four-handed delivery system with both an experienced operator and an assistant may have allowed a considerably well-ordered application with similar time placement, whereas Alhareky’s study was conducted without an assistant.

The difference in our results may also be due to the application of sealants immediately after etching, whereas the two previous studies applied adhesive and drying agents. Neither agent was applied in this study due to applying the sealant strictly according to the manufacturer’s instructions.

Although an insignificant difference in chair time was noted, significant findings were observed in this clinical study. Regarding participant preference, the results indicated that IS was considered significantly noisier than CRI and RDI. This result coincides with Collette’s and Alhareky’s findings and may be due to the continuous suctioning property of IS.

It was found that CRI stretched the participants’ mouths the least and was the most comfortable, followed by IS and RDI. These findings were expected for CRI due to the softness of the material and its small size, allowing it to be placed without causing much discomfort. However, the finding that RDI caused stretching more than IS may be due to the presence of the clamp, stretching the rubber dam over it causing retraction of the cheek and tongue, and adapting the dam to the frame, which facilitated more rubber dam contact with the perioral area. Additionally, the presence of a bite block component in the IS allowed the patient to rest their jaw by biting on the component rather than continuously asking the patient to open wide as in RDI, where an external bite block is not used.

Regarding future preference, the majority of participants significantly preferred to use CRI in the future and least preferred RDI, which may be due to the ease of CRI and the discomfort associated with RDI components, which may impinge on the gingiva, buccal mucosa, and tongue, causing discomfort and irritation. Isolite systems may not have been the most preferred technique due to some discomfort caused by the silicone and continuous suction. These findings are in keeping with Alhareky’s results, where IS was preferred to RDI; however, Collette’s findings were contradictory to this study, where IS showed no significant difference in preference when compared to CRI. These observed variations among the results could be attributed to the differences in the number and skills of the operators applying the PFS and the use of three isolation techniques rather than only two.

Although RDI is considered the recommended isolation procedure for PFS placement [12], this clinical trial supports the results of previous studies indicating RDI to be the most disliked technique [14,15]; however, to determine whether IS or CRI should be considered a recommended isolation technique for sealant placement in the same manner as RDI, a clinical study on sealant retention comparing the three isolation techniques is essential. All participants in this study will be followed up in the future to assess the PFS retention rate.

Finally, previous studies have compared IS to either RDI or CRI alone but not all three combined, whereas this clinical trial is considered one of the first to implement this comparison. The comparison of three isolation techniques in a single patient may have permitted the participants to experience more options, which could improve patient satisfaction, hence improving our practice as pediatric dentists.

## 5. Conclusions

It was concluded that there was no statistically significant difference in PFS placement time among the three isolation techniques. However, cotton roll isolation was the most preferred technique, while rubber dam isolation was the least preferred technique by the participants.

## Figures and Tables

**Figure 1 children-08-00444-f001:**
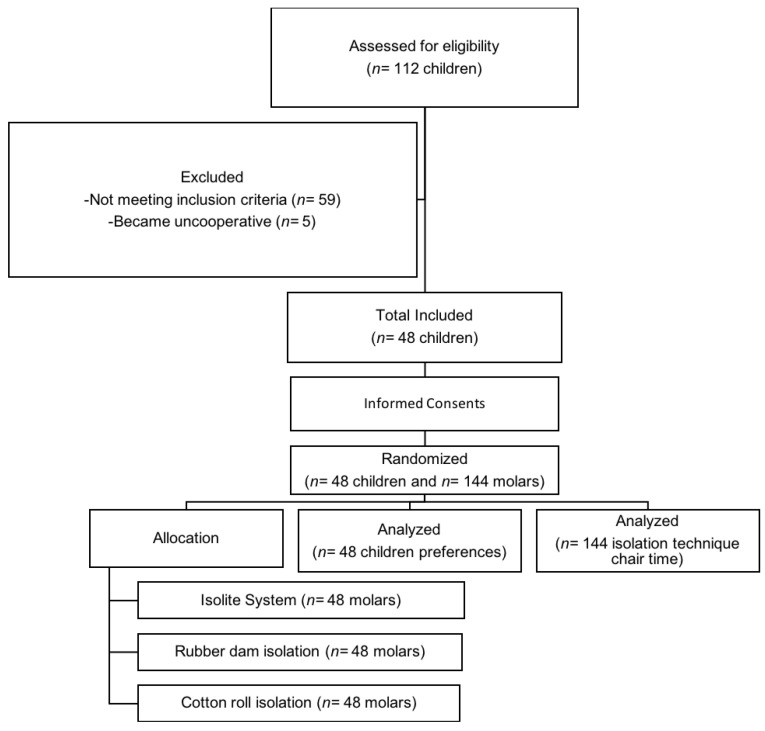
Flow diagram of the study following CONSORT recommendation.

**Figure 2 children-08-00444-f002:**
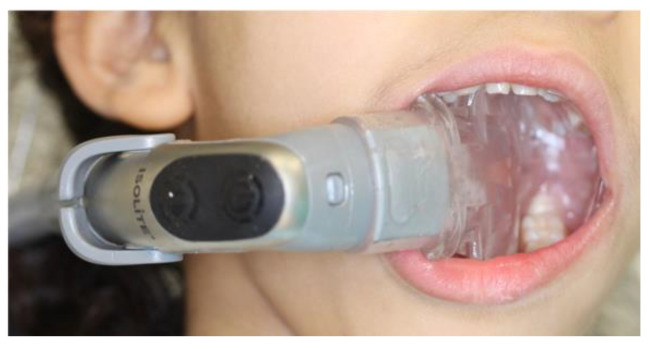
The Isolite System.

**Figure 3 children-08-00444-f003:**
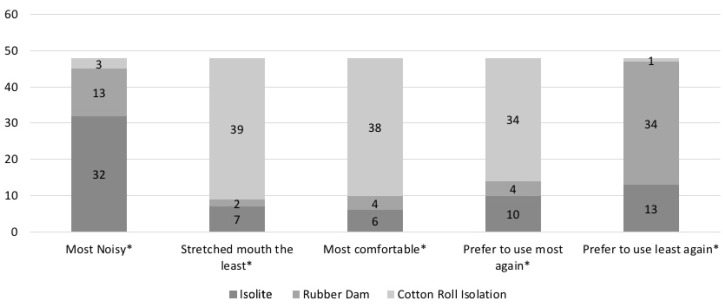
Patient preference for isolation techniques. * Differences are statistically significant (Chi-square test).

**Table 1 children-08-00444-t001:** Comparison of time taken to place the sealant.

	Mean	Std. Deviation	F *	*p* * Value
Isolite	248.15	85.40	0.151	0.700
Rubber Dam	255.90	99.52		
Cotton Roll	243.29	68.22		

F * calculated using the repeated measures ANOVA; *p* * Value for repeated measures ANOVA.

**Table 2 children-08-00444-t002:** Differences in time of placement of sealant between the maxillary and mandibular arches.

	Arch of Placement	N	Mean	Std. Deviation	t *	Sig
Isolite	Maxillary	24	256.46	97.11	0.670	0.507
	Mandibular	24	239.83	72.99		
Rubber Dam	Maxillary	24	273.71	119.51	1.247	0.219
	Mandibular	24	238.08	72.77		
Cotton Roll	Maxillary	24	238.67	65.58	−0.446	0.644
	Mandibular	24	247.92	71.87		

t * calculated using the independent samples; Sig level of significance *p* < 0.05.

**Table 3 children-08-00444-t003:** Correlation between time taken for sealant placement and the age of the child according to type of isolation.

Type of Isolation			Time for Placement
Isolite	Age	Pearson Correlation	−0.255
		Sig. (2-tailed)	0.080
		N	48
Rubber Dam	Age	Pearson Correlation	−0.139
		Sig. (2-tailed)	0.346
		N	48
Cotton Roll	Age	Pearson Correlation	−0.082
		Sig. (2-tailed)	0.579
		N	48

## Data Availability

The data presented in this study are available on request from the corresponding author. The data are not publicly available.

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
