# Peer review of "Comparison of Fissure Sealant Chair Time and Patients’ Preference Using Three Different Isolation Techniques"

_children, 2021, doi:10.3390/children8060444_

Round 1

Reviewer 1 Report

The present article is finished with the remark: to determine whether IS or CRI should be considered a recommended isolation technique for sealant placement in the same manner as RDI, a clinical study on sealant retention comparing the three isolation techniques is essential. All participants in this study will be followed up in the future to assess the PFS retention rate.

This intention is very important in evaluating the study and ought to be clarified from the beginning. 

Reviewer 2 Report

The paper itself is well written and documented, showing a great effort from the authors.

The topic sounds original and with an interesting clinical meaning.

I would make only the following few mentions:

Line 115: How did you perform the randomization?

Line 147: It could be useful to add a figure of the Isolite system

Otherwise, the article is good.
